# A Simple Method for Quantification of Polyhydroxybutyrate and Polylactic Acid Micro-Bioplastics in Soils by Evolved Gas Analysis

**DOI:** 10.3390/molecules27061898

**Published:** 2022-03-15

**Authors:** Jakub Fojt, Ivana Románeková, Petra Procházková, Jan David, Martin Brtnický, Jiří Kučerík

**Affiliations:** 1Institute of Chemistry and Technology of Environmental Protection, Faculty of Chemistry, Brno University of Technology, Purkynova 118, 612 00 Brno, Czech Republic; ivana.romanekova@vut.cz (I.R.); petra.prochazkova1@vut.cz (P.P.); martin.brtnicky@seznam.cz (M.B.); kucerik@fch.vut.cz (J.K.); 2Group of Environmental and Soil Chemistry, iES Landau, Institute for Environmental Sciences, University of Koblenz–Landau, Fortstraße 7, 76829 Landau in der Pfalz, Germany; david@uni-landau.de; 3Department of Agrochemistry, Soil Science, Microbiology and Plant Nutrition, Faculty of AgriSciences, Mendel University in Brno, Zemedelska 3, 613 00 Brno, Czech Republic

**Keywords:** microplastics, biodegradable plastics, micro-bioplastics, soil, evolved gas analysis

## Abstract

Conventional plastics are being slowly replaced by biodegradable ones to prevent plastic pollution. However, in the natural environment, the biodegradation of plastics is usually slow or incomplete due to unfavorable conditions and leads to faster micro-bioplastic formation. Many analytical methods were developed to determine microplastics, but micro-bioplastics are still overlooked. This work presents a simple method for determining poly-3-hydroxybutyrate and polylactic acid micro-bioplastics in soil based on the thermogravimetry–mass spectrometry analysis of low molecular gases evolved during pyrolysis. For the method development, model soils containing different soil organic carbon contents were spiked with micro-bioplastics. Specific gaseous pyrolysis products of the analytes were identified, while the ratio of their amounts appeared to be constant above the level of detection of the suggested method. The constant ratio was explained as a lower soil influence on the evolution of the gaseous product, and it was suggested as an additional identification parameter. The advantages of the presented method are no sample pretreatment, presumably no need for an internal standard, low temperature needed for the transfer of gaseous products and the possibility of using its principles with other, cheaper detectors. The method can find application in the verification of biodegradation tests and in the monitoring of soils after the application of biodegradable products.

## 1. Introduction

Due to the steadily increasing usage of plastic materials, microplastic particles (microplastics, MPs) of 1–1000 µm in dimension are entering all environmental compartments [1,2,3,4,5]. The most affected environmental compartment is soil, where the microplastic concentrations were estimated to be up to 23 times higher than in aquatic ecosystems [6]. A major source of microplastics in soil is agriculture, where they are formed due to the deterioration of mulch foils, tunnels, holders and other plastic structures [7]. Microplastics can also enter soil via the application of compost or wastewater sludge [8,9,10,11]. Another source of MPs in soil is the abrasion of particles from tires (rubber particles) and subsequent atmospheric deposition [12,13].

For this reason, there is growing concern about the environmental impact of microplastics on the soil biota. For example, it was found that poly(ethylene) (PE) microplastics in soil negatively affect spermatogenesis in earthworms [14], and that springtail mobility in soils is reduced by microbead and microfiber contamination [15]. Moreover, negative effects of MPs were also observed on plant root growth [16,17]. Soil microbiota is also negatively affected by MPs [18], either by the inactivation of soil enzymes [19] or by the enrichment of certain strains on the surface of microplastics, which shifts soil functional properties [20]. These effects are complex and not yet fully understood.

The effects on the abiotic components of soil are not understood either [21]. MPs, when present in soil, may reduce the water stability of soil aggregates and change the bulk density [22] or soil density in the rhizosphere [20]; they may also reduce >30 µm soil pore volume [23] or cause soil drying and cracking [24]. Furthermore, all microplastics discursively increase total soil organic carbon, thus biasing soil analysis results [25].

Bioplastics were introduced to prevent overall plastic contamination. Bioplastics can be degraded in the environment by micro-organisms to water and carbon dioxide in aerobic degradation or to methane, carbon dioxide and water in anaerobic degradation [26]. For quick and complete degradation, suitable conditions are required (count of micro-organisms, appropriate pH, temperature, humidity, soil particle size, etc.), which are rarely met in natural environments. Thus, frequently, only the accelerated breakdown into smaller particles of bio-microplastic particles (BMPs) occurs [27,28]. The source of BMPs in soil may then be agriculture through the imperfect biodegradation of slow-release bio-based or -coated fertilizers, mulching foils and the application of contaminated compost [29,30,31]. Soil contamination can also occur through consumers’ improper management of bioplastic waste, such as littering or home composting [32].

Currently, most of the world’s scientists focus on the effects of conventional microplastics on the environment. However, BMP effects on different ecosystems are overlooked, and there is no information on whether replacing conventional plastics with biodegradable ones has more benefits than drawbacks [33]. Ecotoxicological tests on *Daphnia magna*, lettuce and tomato plants showed that BMPs have similar or even worse negative environmental effects than conventional microplastics [34,35,36]. However, on the adsorption of pollutants onto the surface of BMPs and conventional microplastics, i.e., on whether BMPs have comparable or even worse properties than conventional microplastics, very few studies have been published so far [37].

Not all the available assays developed to determine conventional microplastics can be used to determine BMPs in soil. Aggressive reagents that could irreversibly alter the properties and BMP content of the sample should be avoided. Grubelni et al. developed a method to analyze poly(hydroxybutyrate) (PHB) MPs in soil, using depolymerization followed by the determination of the crotonic acid and propene products using liquid chromatography coupled with mass spectrometry (LC/MS) [38]. Low recovery (30–40%) and the requirement of a pure sample make this method unsuitable for environmental analyses. Arcos-Hernandez et al. used Fourier transform infrared spectrometry (FTIR) to determine the PHB content in bacterial cells [39], and Krishnan et al. extracted PHB into chloroform, with PHB being decomposed upon the addition of sulfuric acid, yielding brown crotonic acid, which was subsequently determined spectrophotometrically [40]. Both methods require sample pretreatment, complicating the case of bioplastics in soil. Pyrolysis-based techniques are a promising way to avoid any manipulation of the samples. The pyrolysis products of both PHB and PLA bioplastics are well defined [41,42,43]. Pyrolysis coupled with gas chromatography and mass spectrometry (Py-GC/MS) was used to characterize the composition of polyhydroxyalkanoates produced by bacteria, but it was never applied to determine bioplastics in soil [44].

Analytical approaches for the determination of conventional microplastics are inspirations for the determination of BMPs [27]. For microplastic determination, the studied soil is usually dried [45]; high drying temperatures are not recommended, because they cause changes in the structural properties of the polymers [46]. Since the soil matrix contains large amounts of interfering organic matter, sample purification and analyte extraction are necessary for most of the methods [47]. The simplest and cheapest sample pretreatment method is sieving, which can remove larger particles from the analyzed soil [48]. Suitable density fractionations with water/saturated salt solutions are used to separate microplastics from the solid matrix [47,49,50]. The organic component that these processes cannot remove must then be decomposed, but such decomposition may be performed only if the analytes are resistant to the digestion agent [51]. Catalytic wet oxidations with various reagents are generally used for this purpose [52].

A suitable analytical method for determining microplastics in soil must apply to MP sizes and to microplastics with different compositions, properties, shapes, ages and additive contents [53]. Destructive and non-destructive methods can be used to quantify microplastics in soil. Thermal analysis combined with mass spectrometry (MS) is a typical representative of the former. In this technique, the sample is thermally decomposed without air, and the gaseous products of this reaction are analyzed by MS (pyrolysis directly coupled with gas chromatography with mass spectrometric detection (Py-GC/MS)) [54]. E. Käppler et al. used Py-GC/MS to determine PE, PS, poly(ethylene terephtalate) (PET) and PP microplastics and microfibers in river sediments [55]. Thermogravimetry (TG) can also be used for pyrolysis purposes, as shown by David et al., to determine PET microplastics in soil [56]. The second approach is thermoextractive desorption followed by GC/MS. Here, thermal degradation is carried out in an inert atmosphere; then, its products pass through a solid-phase adsorption bar and are followingly thermally desorbed into the GC/MS detector. This method is suitable for larger sample volumes and can separate the degradation products evolved from labile soil organic matter molecules. Dümichen et al. determined PP, PE and PS microplastics in residues from fermentation tanks and PE, PS and PET microplastics in river sediments by thermal desorption gas chromatography with mass detection (GC/MS) [57,58], while Goßmann et al. used this technique to determine microplastics resulting from tire abrasion [59].

Microplastics in soil can also be extracted with a suitable solvent and then determined in solution by means of high-performance liquid chromatography (HPLC) or quantitative proton nuclear magnetic resonance (q-^1^H-NMR). To quantify PET and polycarbonates, Wang et al. used alkaline depolymerization followed by liquid chromatography with mass spectrometric detection LC/MS [60]. The disadvantage of this technique is incomplete depolymerization, which leads to high uncertainty in the analyses. Nelson et al. extracted the micro-bioplastics formed from poly(butylene adipate-co-terephthalate) (PBAT) mulch foil into deuterated chloroform using various methods and then determined them by q-^1^H-NMR [61]. According to the authors, this analysis is simple, does not require any sample pretreatment and is suitable for the fast and precise determination of microplastics in soil.

The most used non-destructive methods for the determination of microplastics in soil are vibrational spectroscopy techniques. Their advantage is the possibility of determining the number and shape of the particles, though they are limited by the needs of a blank and of drying the sample [53]. Fourier transform infrared with attenuated total reflection (FTIR-ATR) is the most used method to determine microplastics in soil, since it allows one to analyze larger particles and does not require any special sample pretreatment [54]. Raman spectrometry can be used as a complementary method to FTIR. The advantage of this technique is that it can analyze samples containing water, which is a strong interferent in FTIR. Nevertheless, interference occurs due to the inorganic and organic components contained in soil. Therefore, it is necessary to remove the interfering components of the sample using the pretreatment methods mentioned above, which means that neither of the two methods is suitable for the determination of micro-bioplastics in soil [53].

This work aims to develop a simple, rapid, robust, solvent-free, scalable and cheap method that would be suitable for detecting and quantifying PHB and PLA micro-bioplastics in soil, which would be suitable for the assessment of the contamination of agricultural soils and the verification of biodegradation experiments. The method of choice is thermogravimetry coupled with mass spectrometry (TG-MS), which was already demonstrated to determine conventional microplastics with no special sample pretreatment [56].

## 2. Materials and Methods

### 2.1. Materials Used for Experiments

Two plastics with the greatest future potential, PHB and poly(lactic acid) (PLA), were used to develop this method. PHB was in powder form (most particles between 64 and 125 µm) from Y1000P (TianAn Biologic Materials, Ningbo City, China). Ingeo 4060D PLA from NatureWorks (Minnetonka, MN, USA) was obtained in granules. To obtain microplastics, the granules were cooled with liquid nitrogen and ground with a shear mill to particles smaller than 1 mm [62].

### 2.2. Soils Used for Experiments

In total, five soils were used. The first soil was LUFA 2.2 soil (LUFA, Speyer, Germany) (see Table 1 for properties). Next, two soils collected in Siberia in 2010 were used for analyses with PHB (Table 1). Last, soils from the vicinity of the villages of Šaratice and Postoupky in South Moravian Region (Czech Republic) were used for PLA analyses (Table 1). After being received, all soils were air-dried and sieved through a 2 mm sieve.

### 2.3. Preparation of Calibration Mixtures

Calibration mixtures of soil and MBPs were prepared directly in 85 μL alumina TGA crucibles (NETZSCH, Selb, Germany) for better homogenization, using Mettler AE240 analytical balances. A total of 40 mg of soil was always mixed with MBPs to achieve the desired wt% concentration. For PHB, a concentration series of calibration mixtures was prepared in LUFA with PHB content from 0.19 to 3.04%; in HS5 soil, from 0.9 to 2.95%; and in soil HS45, from 0.41 to 3.08%. For PLA, a concentration series of calibration mixtures was prepared in LUFA with PLA content from 0.09 to 5.00%; in P84 soil, from 0.21 to 5.30%; and in P185 soil from 0.05 to 5.09%.

### 2.4. Evolved Gas Analysis System

The analysis was performed using a Netzsch Jupiter STA 449 F1 thermal analyzer (NETZSCH, Selb, Germany) connected by a 3 m long heated capillary with an internal diameter of 60 µm to an Agilent 5977B MS MSD (Agilent Technologies, Santa Clara, CA, USA). The ionization source used for the experiments was electron impact ionization, and the data were recorded in selected ion monitoring mode. The gaseous products went directly into the MS. Each calibration mixture was measured only once due to the impossibility of preparing an identical mixture in the pan. Instead, multiple calibration mixtures were created to generate calibration curves. The measurements were carried out in an inert argon atmosphere with a 50 mL/min flow rate. The capillary connecting the TG and MS systems was heated to 120 °C to avoid the frequent problem of high molecular-degradation-product sorption and desorption on the capillary (no need to use an internal standard), to achieve the most economical and transferable conditions and to use older and less heated capillaries for the measurements. The PHB measurements were carried out at a 10 K/min heating rate from 36 °C to 500 °C. The PLA measurements were also performed at a 10 K/min heating rate from 36 °C to 560 °C.

### 2.5. Evolved Gas Analysis System

First, pure bioplastics were measured to determine specific pyrolysis products that enabled them to be detected. To determine the *m*/*z* of PHB degradation products (for degradation products and MS fragments, see Appendix A), 2.90 mg of pure PHB was measured first, then 45.51 mg of pure soil was measured as a blank; finally, soil containing 2.95% PHB (containing 1.31 mg of PHB) was measured. To select the suitable *m*/*z* for PHB detection, the blank signal had to be negligible, and the peak maxima for pure PHB, PHB-contaminated soil and PHB mass loss from thermogravimetry (in which case there was a small lag because of the delay in the transfer capillary) had to correspond with each other. These conditions were met for *m*/*z* 68 (Figure 1a), which corresponds to the crotonic acid fragment, and *m*/*z* 86, which corresponds to crotonic acid (Figure 1b).

To determine the *m*/*z* of PLA degradation products (for degradation products and MS fragments, see Appendix A), 35.14 mg of pure PLA was first measured, then 39.32 mg of pure soil was measured as a blank; finally, soil containing 5.30% PLA (containing 2.21 mg of PLA) was measured. In addition, to select the appropriate *m*/*z* for PLA detection, the blank signal had to be negligible and the peak maxima for pure PLA, PLA-contaminated soil and PLA mass loss from thermogravimetry had to correspond with each other (in this case, there was a small lag because of the delay in the transfer capillary). These conditions were met by *m*/*z* 29 (Figure 2) and *m*/*z* 43, which are acetaldehyde fragments. *m*/*z* 44 was not used for detection, even though it is an abundant ion for acetaldehyde because it is commonly evolved during pyrolysis of other microplastics and organic compounds. The monomer (*m*/*z* 56) and oligomers (*m*/*z* 100 and 128) of PLA observed by Arrieta et al. were not detected, probably due to condensation or sorption of these pyrolysis products in the capillary, as described previously by Dümichen et al. and Schindler et al. [63,64,65].

To avoid the condensation and sorption of the degradation gaseous products in the capillary, the measurements were repeated with 21.24 mg of pure PLA with a capillary transfer temperature of 280 °C. In this experiment, all *m*/*z* missing in the previous experiment were detected.

### 2.6. Evolved Gas Analysis System

The limit of detection (LOD) and limit of quantification (LOQ) were calculated according to German standard DIN 32645 using the R package envalysis v0.3.3 (code publicly available from https://doi.org/cn74, accessed on the 20 November 2021), which was used by [56,66].

## 3. Results and Discussion

### 3.1. Results Obtained at a Low Capillary Temperature

Figure 3 reports exemplary peaks of the soil and PLA calibration mixtures for P185 soil and *m*/*z* 29 and the peaks of the PHB-spiked soils for HS5 soil and *m*/*z* 86. This figure shows that the PLA-spiked-soil peaks were symmetrical, while the PHB-spiked-soil peaks tailed off to higher temperatures. This phenomenon was observed for all PHB-containing soils; probably, this can also be related to the sorption of gaseous degradation products in the capillary. Thus, as shown by the results of the analyses, the soils investigated in this work (and it can be assumed that this is true for mineral soils in general) produced negligible amounts of gases interfering with the signal at lower *m*/*z*, which was stronger and easier to process.

The peaks obtained for PHB-spiked soils were integrated with the temperature interval from 265 to 380 °C, which was from 295 to 410 °C for PLA. Then, the intensities of all signals were normalized to the signal of the calibration mixture with the highest concentration. The resulting parameters of the calibration lines for PHB determination are given in Table 2, and a sample calibration line for HS5 soil and *m*/*z* 68 is shown in Figure 4 (other calibration lines are shown in Appendix A). Similarly, the parameters of the calibration lines for PLA determination in soil are given in Table 3, and a sample calibration line for P185 soil and *m*/*z* 29 is shown in Figure 5 (other calibration lines are shown in Appendix A).

In contrast to ref. [56], where LOQs of 18.40 and 51.00% were achieved without the use of an internal standard and with an internal standard of 1.72 and 6.53% for the soil PET analyses, significantly better results were achieved for PHB even without standardization to an internal standard, with HS45 soil and *m*/*z* 86 performing the worst, with an LOQ of 4.00%. However, for this soil, it was possible to use *m*/*z* 68, thus reducing the LOQ to 1.61%. The results for HS5 soil were comparable for both *m*/*z* 68 (1.56%) and *m*/*z* 86 (1.60%). For the determination of PHB in LUFA soil, the LOQs for both *m*/*z* were also found to be comparable, at 2.82% and 3.00%, respectively. In the case of determination of PLA in soil, the LOQs were higher, which may be because the selected *m*/*z* were not specific to PLA alone but were also released to a lesser extent by the soils themselves. The best result was obtained for P84 soil and *m*/*z* 29, with an LOQ of 1.82%. In addition, a higher LOQ was found in LUFA soil for this MBP, ranging from 2.88 to 3.88%.

### 3.2. Influence of Soil on the Analysis of Micro-Bioplastics

As shown in the previous section, both PLA and PHB provide some typical fragments that can be used qualitatively and quantitatively to analyze these MBPs in soil. Thus, their presence can indicate the presence of MBPs in soil. However, it cannot be ruled out that these gaseous degradation products released during the thermal degradation of PLA (acetaldehyde) may also be released from the soil, and interference between soil and MBP signals may occur. The source of these gases is soil organic matter, from which ions with the same *m*/*z* as those of pure bioplastics may be released. Therefore, soils with a higher proportion of soil organic matter, which may overlay lower concentrations of micro-bioplastics, could become a problem. The solution to this potential problem is the observation, as the results of the previous chapter suggest, that the gaseous products from pure MBPs always leave in a constant stoichiometric ratio, unlike in soils where this ratio is different.

To demonstrate this phenomenon, the organic matter or organic carbon content of the tested soils had to be considered. The organic carbon in the studied soils ranged from 0.59 ± 0.24% for P84 soil to 6.70 ± 0.25% for HS45 soil (Table 1). To assess the effect of soil on PLA analyses, the ratio of the areas of all ion peaks at *m*/*z* 43 and *m*/*z* 29 was calculated and plotted versus concentration in a graph (Figure 6). This figure shows that, at low MBP/soil organic matter ratios, the signal was significantly affected by gases released from the soil. As the concentration (i.e., MBP/soil organic matter ratio) increased, the peak-area ratio was constant at concentrations from approximately 0.5 to 1%, which was correlated with the detection limit. In contrast, no relationships were observed between soil organic matter content and the blank peak-area ratio deviation from the peak-area ratio at high soil PLA concentrations.

A similar effect was observed for the PHB ion peak ratio at *m*/*z* 68 and 86 (Figure 7). This suggests that the composition of soil organic matter was more important than its concentration for the studied soils. We can also infer that the thermal decomposition of these soils results in a negligible release of detection gases for both bioplastics, which only affects the analyses below the detection limits; thus, all the selected ions can be used for the detection of these bioplastics.

Our results show that microplastics can be analyzed even by using lower molecular weights of pyrolysis gases using TG-MS. Furthermore, there was an attempt to keep the capillary at the lowest possible temperature, which minimized the possibility of adsorption of small molecules on the column. The analysis based on lower molecular weights increases the possibility to scale the analysis up. In other words, this approach can be used in analyses with other techniques equipped with cheaper detectors, such as the flame ionization detector (GC-FID). A major weakness of microplastic analyses of natural soils is the sampling of the soil, especially if it is contaminated with smaller plastic particles. Most methods are based on soil sampling, homogenization and subsequent quartering, thus reducing the sample volume. In the case of soil and similar substrates, such as sludge, this can lead to under- or overestimating the results. Therefore, a batch method is needed for some applications. The results reported here suggest the possibility of conducting the analyses in a large-scale pyrolizer (i.e., pyrolyzing large amounts of samples) connected to a simple detector, thus enabling the analysis of small molecules evolved from the contaminated soil to be conducted. In addition, the use of a constant ratio between typical gases avoids the use of internal standards, which can be problematic when using batch methods.

### 3.3. Results Obtained at a High Capillary Temperature

To demonstrate the suitability and advantage of the proposed approach, i.e., low capillary temperature and low molecular weight, we also present the data obtained in the experiment with high capillary temperature. A higher capillary transfer temperature could indeed result in a lower LOQ for the determination of MBPs in soil. This could be due to a better detection of higher specific *m*/*z*, as the direct condensation and sorption of gaseous products in the capillary would not occur. Increasing the capillary temperature could also reduce peak tailing, as observed for PHB (Figure 3).

The results show that, although these *m*/*z* are specific to PLA and would be useful for samples containing higher amounts of organic matter, the signal at *m*/*z* 100 and 128 was not intense enough for the analysis of real samples (Figure 8). Although the signal at *m*/*z* 56 (Figure 9) from the PLA dimer was more intense, there was signal splitting, and the main peak was not separated from the other two peaks [67]. The last peak ended at a temperature higher than 550 °C, which increased the analysis time and was instrumentally more demanding. In addition, it extended the peaks over a larger temperature range, which is generally problematic for quantitative analyses. Moreover, even in this case, sorption onto the column cannot be excluded and further temperature increases could lead to secondary reactions. In addition, the upscaling of this analysis discussed in the previous chapter would be technologically problematic due to the need to maintain the equipment at the desired temperature.

### 3.4. Other TG-MS Measurement Alternatives

Another alternative that could improve the LOD and LOQ determination of MBPs in soil would be using a system where the TG detector is coupled directly to the MS detector without using a capillary [65]. Although this method requires minimal pretreatment of the sample prior to measurement, which is necessary for analyzing MBPs, its major drawback is the small amount of sample used for the analyses. Thus, this method needs to ensure thorough homogenization of real samples and higher replication of measurements to obtain representative results. It would also be problematic to analyze soils with larger bioplastic particles. This problem could be solved by using a system that can measure larger sample volumes.

The determination of both bioplastics in soils and other matrices (e.g., activated sludge and compost) rich in organic matter remains an analytical challenge, because the detection *m*/*z* for PLA can be strongly released from the matrix itself during analysis. For PHB, it would be necessary to distinguish anthropogenic PHB from natural PHB, which is primarily produced by bacteria in activated sludge as a carbon and energy storage polymer [68].

## 4. Conclusions

In this work, we present, for the first time, a method for the determination of PHB and PLA in soil using TG-MS. The method is inspired by the work of [56], where TG–MS were used for the quantitative analysis of PET microplastics in standard LUFA soil. This method uses a capillary heated to a lower temperature, and lower *m*/*z* are used for detection; therefore, it can be adapted to cheaper and simpler TG-MS or Py-MS. In this work, we show that similar or better results can be achieved in LUFA soil, and that the method can be used for analyses of real soils.

This work aims to develop a simple and, possibly, scalable method to analyze residual MBPs after biodegradation experiments. Since the recommended concentration of bioplastics in biodegradation experiments according to ISO 17556 is 0.1%, the LOD and the LOQ are not sufficient to detect residual bioplastics. The possible inclusion of this technique in the verification of biodegradation tests would need to be refined by using an internal standard, by analyzing a larger volume of the sample, or by using a larger number of samples in the biodegradation experiments. As bioplastics currently account for only a fraction of the world’s plastic production, and thus have not yet entered the environment in large quantities, it is not possible to estimate the environmentally relevant concentrations; thus, it is not possible to assess the suitability of the method for analyzing real samples. On the other hand, the development of bioplastic fertilizers and bioplastic mulch films may lead to the accumulation of bioplastics in agricultural soils; thus, the number of bioplastics in soil could be detectable by this method [29,69].

## Figures and Tables

**Figure 1 molecules-27-01898-f001:**
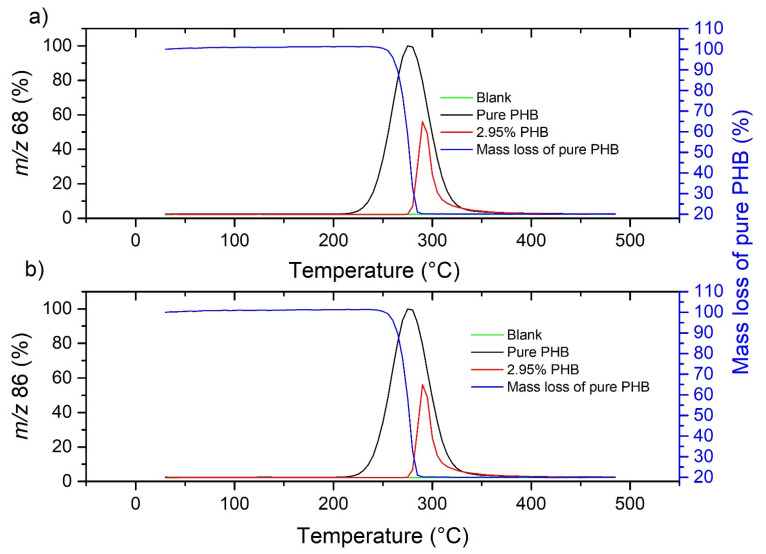
Plots of the peaks at *m*/*z* 68 (**a**) and 86 (**b**) of blank, pure PHB and HS5 soil containing 2.95% PHB suitable for PHB detection.

**Figure 2 molecules-27-01898-f002:**
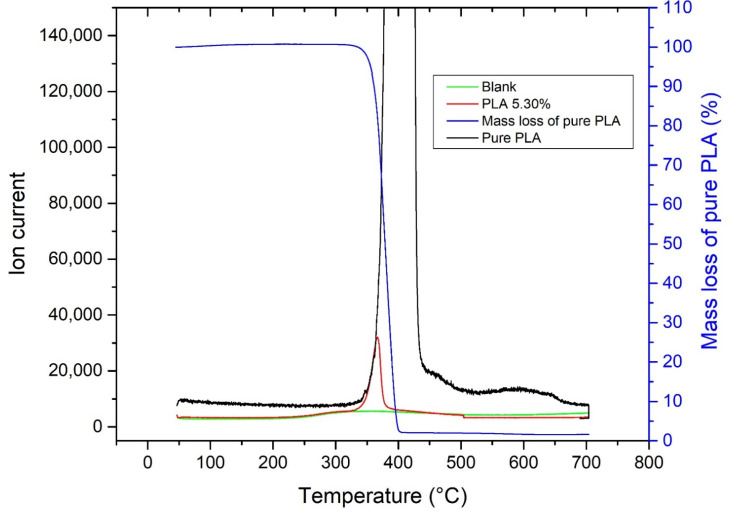
The plot of the peaks at *m*/*z* 29 of blank, pure PLA and P185 soil containing 5.30% PLA suitable for PLA detection.

**Figure 3 molecules-27-01898-f003:**
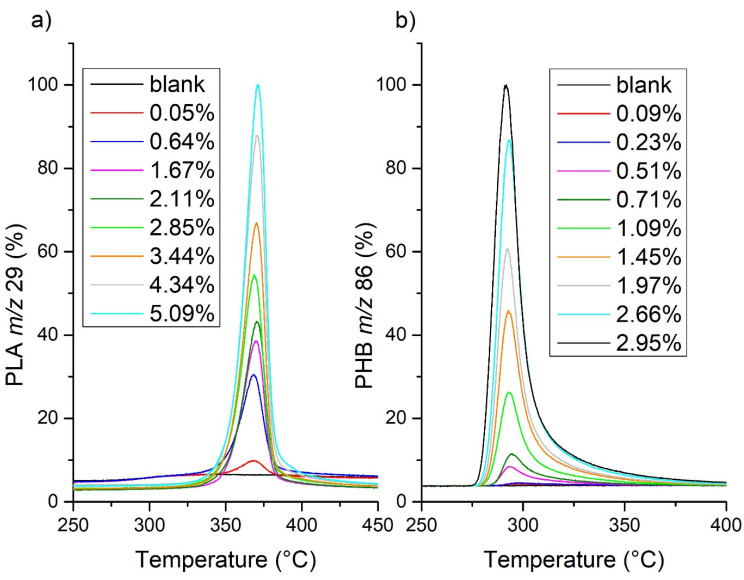
Example PLA (**a**) calibration mixture peaks for soil P185 and *m*/*z* 29 and PHB (**b**) calibration mixture peaks for soil HS5 and *m*/*z* 86.

**Figure 4 molecules-27-01898-f004:**
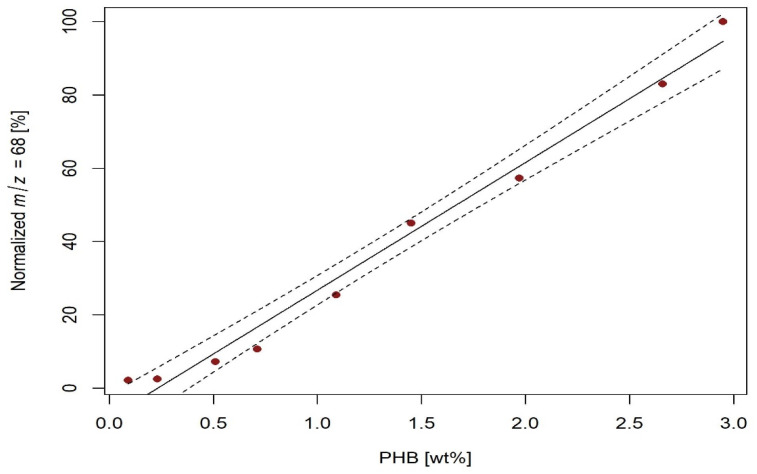
Calibration line for determination of PHB in HS5 soil by detection of degradation products with *m/z* 68; dashed line indicates 95% confidence interval.

**Figure 5 molecules-27-01898-f005:**
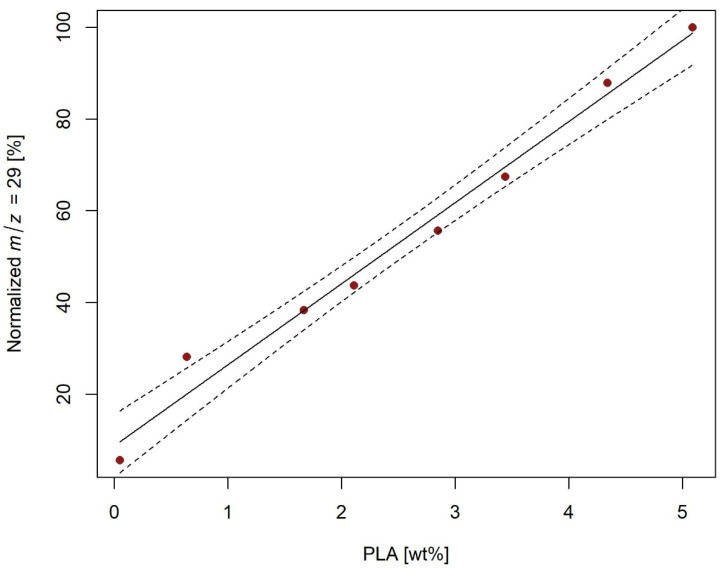
Calibration line for determination of PLA in P185 soil by detection of degradation products with *m*/*z* 29; dashed line indicates 95% confidence interval.

**Figure 6 molecules-27-01898-f006:**
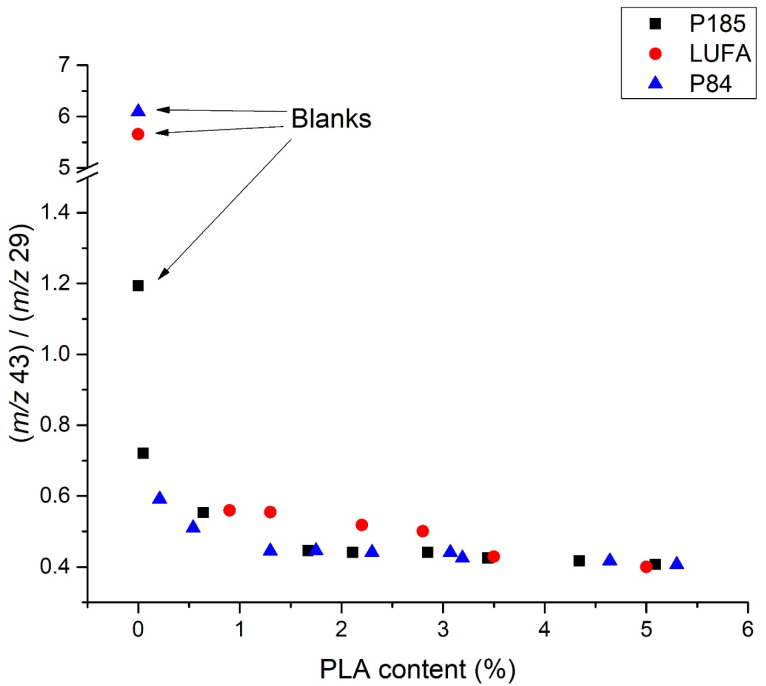
Ratios of PLA peaks of ions with *m*/*z* 43 and 29 in different soils as a function of MBP concentration.

**Figure 7 molecules-27-01898-f007:**
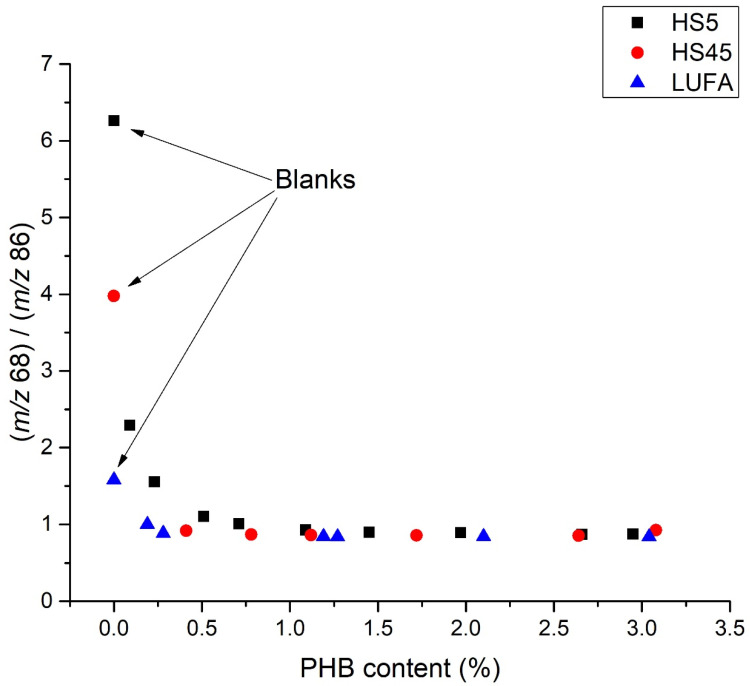
Ratios of PHB peaks of ions with *m*/*z* 68 and 86 in different soils as a function of MBP concentration.

**Figure 8 molecules-27-01898-f008:**
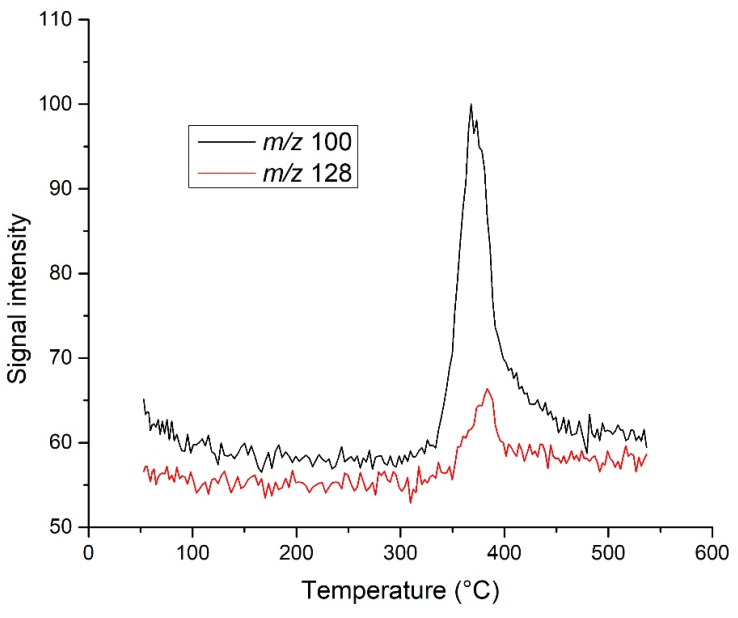
Comparison of peaks at *m*/*z* 100 and 128 obtained with the capillary heated at 280 °C.

**Figure 9 molecules-27-01898-f009:**
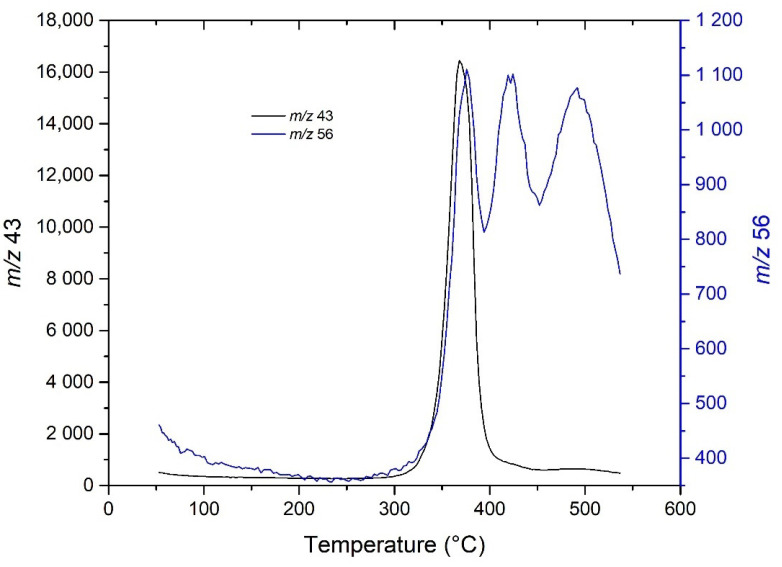
Peaks at *m*/*z* 56 obtained with the capillary heated to 280 °C. The peak of the signal at *m*/*z* 43 from the same experiment is also shown for comparison (n.b., the two peaks have different y-axes).

**Table 1 molecules-27-01898-t001:** Properties of soils used for experiments.

Soil	Soil Type	C_org_ (%)	Sampling Location
P84	Luvic cambisol	0.59 ± 0.24	49°31′81.36″ N
			17°35′88.31″ E
P185	Phaeozem	1.47 ± 0.69	49°11′74.64″ N
			16°80′35.13″ E
LUFA 2.2	Loamy sand (lS)	1.77 ± 0.56	n.a.
HS5	Protocalcic chernozem	4.88 ± 0.05	55°33′12.96″ N
			84°08′06.60″ E
HS45	Albic Luvisol	6.70 ± 0.25	56°51′33.43″ N
			83°04′26.71″ E

**Table 2 molecules-27-01898-t002:** Parameters of calibration lines for the determination of PHB micro-bioplastics in standard LUFA soil and two real soils by detection of degradation gas products with *m*/*z* 68 and 86.

Soil	*m*/*z*	Slope	Intercept	Adj. R^2^	RSE (%)	LOD (%)	LOQ (%)
HS5	68	8166	−1439	0.9822	3.249	0.48	1.56
	86	9491	−2068	0.9811	3.372	0.49	1.60
HS45	68	5126	159	0.9897	0.224	0.52	1.61
	86	5773	250	0.9662	0.425	0.95	4.00
LUFA	68	9630	394	0.9775	0.888	0.75	2.82
	86	11,529	303	0.9761	0.921	0.77	3.00

**Table 3 molecules-27-01898-t003:** Parameters of calibration lines for the determination of PLA micro-bioplastics in standard LUFA soil and two real soils by detection of degradation gas products with *m*/*z* 29 and 43.

Soil	*m*/*z*	Slope	Intercept	Adj. R^2^	RSE (%)	LOD (%)	LOQ (%)
P84	29	176,237	58,873	0.9808	1.977	0.80	2.55
	43	72,464	30,106	0.9913	1.104	0.57	1.82
P185	29	180,984	65,753	0.9809	5.457	0.91	2.88
	43	69,949	50,891	0.958	9.513	1.37	4.72
LUFA	29	188,508	−3302	0.9839	0.388	0.96	2.88
	43	71,715	35,492	0.975	0.428	1.20	3.76

## Data Availability

The data presented in this study are available upon request from the corresponding author.

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
