# Peer review of "A Simple Method for Quantification of Polyhydroxybutyrate and Polylactic Acid Micro-Bioplastics in Soils by Evolved Gas Analysis"

_molecules, 2022, doi:10.3390/molecules27061898_

Round 1

Reviewer 1 Report

A simple method for quantification of polyhydroxybutyrate and polylactic acid micro-bioplastics in soils by evolved gas analysis

This manuscript presents a simple method for determining polyhydroxibutarate and polylactic acid in soil focused on mico-bioplastics topic using thermogravimetry-mass spectrometry. The authors test this technique in different soils which change the organic carbon content and spiked with the analytes. The manuscript approach focused on analytical chemistry.

In general, the whole study is properly constructed and executed. The results were clearly represented by means of figures and tables that were thoroughly discussed and correlated with existing literature data.

However, I have some suggestions to included before the manuscript is accepted to publish in the journal:

  • The introduction is extensive, I understand that the authors need to explain the global situation of microplastics, however, the title of the manuscript says “A Simple Method for Quantification…”, it is very important to included similar studies that use thermogravimetry-mass spectrometry and mention which are the advantages comparing with RMN and HPLC that the authors mention in the introduction. Finally, synthesize the environmental information. With this context, I suggest starting the introduction with the Environmental topic, then the global situation of microplastic, analytes, and finally focused on analytical techniques. Please consider that in mind this information.
  • Please in table 1, order the table according to with the %C first with the less and finally with the more %C.
  • In section 2.4 Envolved gas analysis system, in gas chromatography to mass spectrometry, What gas was used as a mobile phase? Please clarify this information and describe the analytical method step by step, because the goal of this manuscript is to inform a new method.
  • In Section 2.5, when the authors determine the m/z of PHB, What ionization source was used? Electronic impact? – Was the Mass spectrometer operated in Full Ion or SIM? Please, again, clarify step by step this information.
  • To understand this information on how the authors select the m/z 68 ions, I suggest attaching a figure with the PHB structure and the possible fragmentation pathway and highlighting which is the m/z 68 fragment.
  • Please, complete the information of PLA as I solicited the PHB information.
  • The validation method and discussion of the results were thorough, please only some sentences need a few English reviews.

Reviewer 2 Report

Although the manuscript does not require correction of methodological errors, I suggest its major revision before a possible acceptance. I have the following comments:

  1. The work concerns micro-bioplastic, whereas most of the introduction section is devoted to microplastic (lines 30-142, 48 references). In my opinion this part should be substantially shortened.
  2. Some papers concerning pyrolysis of polylactic acid or polyhydroxybutyrate should be added and briefly discussed (e.g.:

https://doi.org/10.1016/j.atmosenv.2010.09.035

https://doi.org/10.1016/0141-3910(96)00102-4 https://doi.org/10.1016/j.ijbiomac.2021.01.108 http://dx.doi.org/10.1016/j.jclepro.2014.07.064)

  1. The manuscript will be more readable if the structures of all analyzed pyrolysis products are included (e.g. in the supporting information).
  2. Acetaldehyde yields abundant molecular ion under EI conditions (m/z 44). Why the authors analyzed only its product ions (m/z 43 and 29)? If the reason is the CO2 background (also m/z 44), it should be explained.
  3. Although I do not feel qualified to judge the English language, some phrases seem to be awkward (e.g. line 31, 238, 323, 334)
  4. What is the origin of the signal at m/z 45 (line 372) ?
  5. Reference style does not meet the journal requirements.
  6. m/z should be in italic.

Round 2

Reviewer 1 Report

All figures low quality, please change figure.

Author Response

Dear Reviewer 1,

thank you for your comment about the low quality of our figures. We have revised all figures and all of them meet the quality requirements for the publication in Molecules. Maybe MS Word or conversion into PDF have altered the quality, but all figures provided during submission are in sufficient quality for publication. 

Reviewer 2 Report

see file attached
